SOFTWARE

# `BAGEL`: Protein engineering via exploration of an energy landscape

Jakub Lála[1], Ayham Al-Saffar[1], Stefano Angioletti-Uberti[1,2,3]*

1 Department of Materials, Imperial College London, London, United Kingdom, 2 Nanograb Ltd., London, United Kingdom, 3 AminoAnalytica Ltd., London, United Kingdom

* sangiole@imperial.ac.uk

## Abstract

Despite recent breakthroughs in deep learning methods for protein design, existing computational pipelines remain rigid, highly specific, and ill-suited for tasks requiring non-differentiable or multi-objective design goals. In this report, we introduce `BAGEL`, a modular, open-source framework for programmable protein engineering, enabling flexible exploration of sequence space through model-agnostic and gradient-free exploration of an energy landscape. `BAGEL` formalizes protein design as the sampling of an energy function, either to optimize (find a global optimum) or to explore a basin of interest (generate diverse candidates). This energy function is composed of user-defined terms capturing geometric constraints, sequence embedding similarities, or structural confidence metrics. `BAGEL` also natively supports multi-state optimization and advanced Monte Carlo techniques, providing researchers with a flexible alternative to fixed-backbone and inverse-folding paradigms common in current design workflows. Moreover, the package seamlessly integrates a wide range of publicly available deep learning protein models, allowing users to rapidly take full advantage of any future improvements in model accuracy and speed. We illustrate the versatility of `BAGEL` on four archetypal applications: designing de novo peptide binders, targeting intrinsically disordered epitopes, selectively binding to species-specific variants, and generating enzyme variants with conserved catalytic sites. By offering a modular, easy-to-use platform to define custom protein design objectives and optimization strategies, `BAGEL` aims to speed up the design of new proteins. Our goal with its release is to democratize protein design, abstracting the process as much as possible from technical implementation details and thereby making it more accessible to the broader scientific community, unlocking untapped potential for innovation in biotechnology and therapeutics.

**Data availability statement:** The initial public release of `BAGEL` v0.1 is available at https://github.com/softnanolab/bagel, including all the templates used in this manuscript to reproduce the results. The exact v0.1.0 release with the compatible templates is available on Zenodo at https://doi.org/10.5281/zenodo.15808839. The accompanying package `boileroom` is also publicly available at https://github.com/softnanolab/boileroom.

**Funding:** This work was supported by the President's PhD scholarship at Imperial College London (to J.L.); the UK Materials and Molecular Modelling Hub, which is partially funded by EPSRC (grant numbers EP/T022213/1, EP/W032260/1, and EP/P020194/1 to S.A.-U.); grants from NVIDIA (to S.A.-U and J.L.); and academic computational grants from Modal (to J.L.). The funders had no role in study design, data collection and analysis, decision to publish, or preparation of the manuscript.

**Competing interests:** I have read the journal's policy and the authors of this manuscript have the following competing interests: Stefano Angioletti-Uberti is co-founder and Chief Scientist for Nanograb Ltd and has equity in the company. He is also a paid consultant and has equity in Aminoanalytica Ltd. Nanograb and Aminoanalytica are two biotech companies working on selective drug delivery and protein design, respectively.

## Author summary

Proteins underpin much of modern biotechnology. To harness them, we need practical ways to computationally design sequences that meet clear goals while remaining stable and functional. Recent deep learning methods have boosted success rates, yet most design pipelines still follow rigid workflows, making it difficult to specify non-differentiable goals, or constraints spanning multiple functional contexts. Here, we introduce BAGEL, a Python package that formalizes protein design as exploration of an energy landscape built from simple, user-defined terms. In practice, this lets a researcher specify straightforward constraints – keep a region confident, avoid sticky surface patches, bring two parts into contact, maintain a catalytic motif – and combine them in one place. A stochastic search then proposes beneficial sequence changes, testing whether the design is evolving in the right direction. We demonstrate this on four use cases: de novo peptide binders, targeting intrinsically disordered epitopes, selective binding across species variants, and enzyme variants that preserve an active site. The ability to compose arbitrary goals, including the use of the ever-expanding suite of protein deep learning models makes BAGEL a tool making programmable protein engineering more accessible, accelerating practical applications in biotechnology and therapeutics.

## 1 Introduction

Over the past decade, advances in protein structure prediction - most notably deep learning-based models such as AlphaFold2 [1], RosettaFold [2] and ESMFold [3] - have transformed the field of protein design [4]. The ability to predict the fold of (almost any) arbitrary sequence with relatively high accuracy enables researchers to effectively explore the vast protein sequence space and find those satisfying a set of desired structural and functional properties, increasing the potential for breakthroughs in therapeutic design, biosensing, and industrial biocatalysis [4–6]. Building on top of these developments, a new generation of computational protein design frameworks has emerged [7]. While successful, many remain tightly coupled to specific multi-step workflows inherited from the "inverse folding paradigm" [8], developed prior to the advent of accurate deep learning-based protein-folding algorithms.

The inverse folding paradigm follows the philosophy *backbone design → sequence painting → folding*. In such a workflow, different algorithms (e.g., hallucination [9] or RFDiffusion [10]) are used to determine specific backbone shapes satisfying a given constraint, for example, formation of an interface with a specific epitope on a target protein. Once the fold of these backbones is determined, an inverse-folding algorithm is applied (e.g., the ProteinMPNN-based models [11–13]) to find the sequences most likely to fold into the backbone shapes. Finally, a protein-folding algorithm is used to check if such sequences indeed fold to the correct structure. This last validation step is necessary because, while a candidate sequence obtained as a result of the second step is the most likely to fold into the target backbone, the latter is not necessarily the most stable structure for that sequence. As a result, many designed sequences are rejected and the process is repeated until one is found that successfully folds

into the target backbone. Alternatively to the inverse folding paradigm, one can train a single deep generative model on the joint sequence-structure distribution so that, when conditioned on a desired functional tag, it proposes a complete sequence-structure design in a one-shot generation setting [14–16]; the resulting design also being filtered with a folding predictor. This second approach often leads to poorly interpretable pipelines, with most of the design process becoming abstracted away inside a large neural network.

In both of the previous approaches, conditioning the protein generation on specific functional properties of interest is not particularly straightforward, or even possible. For example, preventing hydrophobic patches to be located on the surface, requesting the presence of certain folds, or enforcing a given symmetry cannot be easily and explicitly implemented. As a result, many different candidates must be generated, and subsequently screened downstream for both validated folding, and wanted properties. Recently, the authors of the hallucination-based pipeline BindCraft [7] recognized that re-folding validation is the critical step for obtaining successful designs. BindCraft's philosophy, therefore, is to maximize the success rate in this validation stage by using AlphaFold2-Multimer as an oracle to co-design sequence-structure pairs directly during backbone generation - rather than following the traditional multi-step pipeline. Such hallucination-based approach, however, requires all the design properties of the system to be fully differentiable, hindering its generalizability to arbitrary design constraints.

To solve these problems, we introduce `BAGEL` (Biomolecular Algorithm for Guidance in Energy Landscapes), a general-purpose Python package for programmable protein engineering. Our framework enables users to formalize protein design tasks as the optimization of an energy function (a loss) over sequence space. In practice, `BAGEL` provides a set of tools to seamlessly build arbitrary sampling and optimization problems by combining energy terms derived from any metric extractable from a sequence. For instance, these can be derived from structure and folding metrics computed by a deep learning structure predictor, vector embeddings from a protein language model (PLM), or metrics based on sequence alone. More specifically, these energy functions include geometric constraints, similarity to user-defined templates, physical properties, and custom objectives based on structural or functional hypotheses, as well as bulk and local confidence scores, e.g., local pLDDT, pTM or interface PAE (iPAE). While various energy terms have already been implemented in this initial release, `BAGEL`'s modular structure also easily allows a user to define any term that could be calculated from knowledge of a (group of) sequence(s). The flexibility of this abstraction allows our framework to support a wide variety of classical protein engineering problems: from templating specific scaffold motifs for de novo enzymes, through building proteins with desired geometry, to designing protein binders. Crucially, to go beyond the capability of current approaches, we also allow to define loss functions that consider multiple states, which is particularly valuable, for example, in engineering proteins that binds to a specific target, while explicitly maintaining a weak interaction with other specific, non-target proteins.

Our package is inspired by - and builds upon - the seminal work of the former FAIR team at Facebook (now Meta), who developed the suite of ESM projects [3,17], and in particular the paper titled *A high-level programming language for generative protein design* by [18]. In that work, the authors use Monte Carlo (MC) optimization to scan the sequence space in search for optimal sequences solving a prescribed protein engineering problem. In contrast to their work, we change the representation of a protein sequence and its associated loss function by shifting away from a graph-based representation in favour of one based on groups of residues, which we believe allows for a more flexible and user-friendly design interface. Besides this change in representation, among other differences that will be addressed later, our fully modular design framework provides:

- **A Multi-State Formalism.** Native support to specify multi-state problems, highly relevant for complex design tasks of, among others, selective binding, or binder design over multiple target variants (cross-reactivity).
- **Enhanced Energy Formalism.** A unified formalism for expressing arbitrary design constraints as energy penalties, following the convention from molecular simulations where total energy is expressed as a sum of simple, fully customizable N-body terms.

- **A Unified Approach to Optimization and Generative Problems.** By looking at protein design as the exploration of an energy landscape, optimization and candidate generations can be seen as two related variations of the same problem: finding the energy minimum (optimization) as opposed to exploring the sequence space around a certain energy value (generation).
- **Multiple Optimization and Sampling Algorithms.** An extensive and easy-to-extend sampling and optimization back-end supporting different types of advanced MC algorithms to accelerate exploration of the sequence space including simulated tempering, grand-canonical MC, and custom annealing schedules.
- **Flexible Oracle Choice.** A generalized and modular implementation of an oracle, which is defined as any algorithm that provides useful information for protein engineering tasks. This includes, for example, protein-folding models that return structures, protein language models that generate embeddings, or other algorithms (or combinations thereof) that compute relevant physicochemical properties for one or more proteins.
- **Accessibility.** A user-friendly interface that can be seamlessly integrated with other Python workflows, including a serverless implementation of deep learning-based protein models (i.e., oracles) through a standalone package, `boile-room` [19], abstracting away tedious dependency management.

Compared to other, problem-specific end-to-end solutions, `BAGEL` has been built from scratch with generality and modularity in mind, enabling continuous extension of the package with new minimization strategies beyond MC and new energy terms describing protein engineering objectives. Moreover, the package is fully model-agnostic, enabling us to leverage the continous advances in deep learning on proteins in terms of both quality and speed.

In this report, we describe the functionality and design philosophy of `BAGEL` through its algorithm formalism and core class abstractions. We illustrate its utility with four examples: i) simple binder design, ii) design of binders targeting intrinsically disordered epitopes, iii) design of selective binders via multi-state optimization, and iv) enzyme variant generation. Fig 1 summarizes the design workflow and the components of the package, including the applications.

## 2 Design and implementation

### 2.1 Algorithm

The energy landscape we aim to explore is defined by a System $\Omega$ and its associated energy $E_\Omega$. Each System itself is composed of a set of States $\{S\}$, each with its own associated energy function $E_S$ dependent on the set of Chains $\{C\}_S$. A specific Chain $C_i$, indicated with the subscript $i$, is a sequence of amino acid residues $\vec{\sigma}_i$, i.e., a monomeric protein, of length $L_i$. In mathematical form,

$$\Omega = (\{C\}, \{S\}) \tag{1}$$

$$S = (\{C\}_S \subset \{C\}, E_S); \ E_S = f(\{C\}_S) \tag{2}$$

$$C_i = \vec{\sigma}_i \in \mathcal{A}^{L_i}, \tag{3}$$

where $\mathcal{A}$ is the amino acid alphabet. We formalize the protein design problem as the sampling of the energy landscape defined by:

$$E_\Omega = \sum_{S \in \Omega} E_S \tag{4}$$

where a State energy function $E_S = E_S(\{C\}_S)$ is a weighted sum over $N_S$ individual energy terms specific to each State:

$$E_S = \sum_j^{N_S} w_{js} \, \epsilon_{js}(\{C\}_S) \tag{5}$$

PLOS Computational Biology

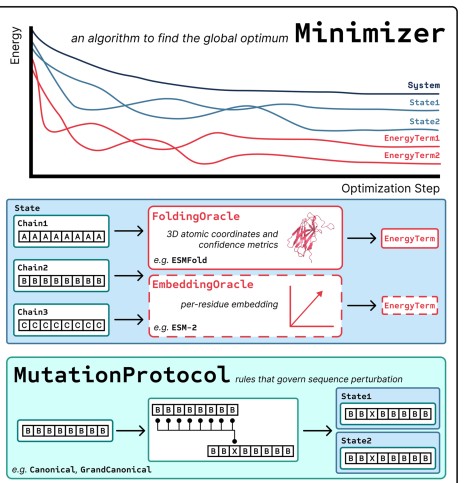
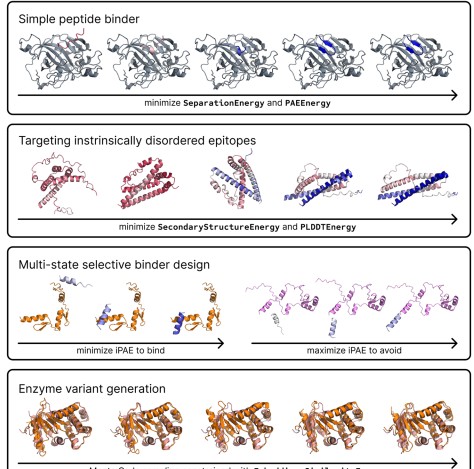

**1) Specify design constraints**   **2) Minimize the System**   **3) Applications**

**Fig 1**. **Conceptual schematic of the `BAGEL` protein design package.** Modular setup of the `System` includes multiple `State`s, each of which has its own set of `EnergyTerm`s defining a specific design objective. Each `State` consists of `Chain`s, that include either mutable or immutable `Residue`s. `EnergyTerm`s come in various flavours; affecting *residue groups* in a 0-body, 1-body, or 2-body fashion, and requiring the ouput of specialized `Oracle`s, such as a `FoldingOracle` or an `EmbeddingOracle`. The latter is symbolized with the dashed outline. A `Minimizer` is used for the case of optimization, shown in this schematic, while another `MonteCarloMinimizer` is used for generative sampling. In this case, the `System`'s energy is optimized with the `Minimizer` through successive sequence perturbations using the `MutationProtocol`. Given the modularity, the package can be applied to a variety of applications, including the four highlighted in the figure detailed below in the report.

The energy terms $\epsilon_{js}$ can be chosen from a plethora of options, shown later in Sect 2.4, but effectively impose a constraint on the design task on a particular State $S$, weighted by $w_{js}$; where $j$ indexes over energy terms, and $s$ over States, as the energy terms are specified for each $S$ independently.

Sequences $C_i$ may be *shared* or *unique* among different States. Let us make an example. Consider the case we want to design a protein $C_0$ that binds to both a protein $C_1$ but also separately to another protein $C_2$. The System $\Omega$ in this case can be composed of two States, let us call them $A$ and $B$. Chain $C_0$ is present in both States, while only $C_1$ is part of State $A$ and only $C_2$ is part of State $B$. We can then optimize $C_0$ by finding a sequence that *at the same time* would be predicted to form a stable multimer for the two independent States $A$ and State $B$.

To calculate the energy terms $\epsilon$, we introduce the general notion of an *oracle*, a black-box predictive computation (i.e., a function) mapping a set of chains $\{C\}_S$ to a useful representation $y_s$. These informative representations $y_s$ are used as inputs to $\epsilon$, thus $\epsilon$ being a function of $y_s$ or $\{C\}_S$ interchangeably. Two important but non-exhaustive examples of classes of oracles are *folding oracles* and *embedding oracles.* Folding oracles are typically deep learning-based structure predictors,

$$f_{\text{fold}} : \{C\}_S \mapsto (\mathbf{X}, \Phi) \tag{6}$$

where $\mathbf{X} \in \mathbb{R}^{\sum_i 3 \times L_i}$ are three-dimensional atomic coordinates and $\Phi$ encodes model confidence metrics. Embedding oracles, in contrast, are language models providing learned high-dimensional features capturing biochemical or evolutionary context:

$$f_{\text{embed}} : \{C\}_S \mapsto (\vec{z}) \tag{7}$$

where $\vec{z}$ is an embedding vector of dimensionality $d = \left(\sum_i L_i\right) \times N_{\text{dim}}$, that is, one embedding vector of dimensionality $N_{\text{dim}}$ per residue in State $S$. Another simpler but general class of an oracle could be a function that maps a (set of) sequence(s)

to a single scalar value $r \in \mathbb{R}$:

$$f_{\text{property}} : \{C\}_S \mapsto r \tag{8}$$

For instance, such oracle could calculate the number of polar residues in a State, or predict the unfolding temperature of a chain.

With the previous definitions, we can define protein engineering as a sampling problem, where the goal is to discover a tuple of sequences among all possible sets $\{C\}$ defining the System that samples a region of the energy landscape for which $E_\Omega(\{C\}_{\text{designed}}) = E_{\text{target}}$. When this target energy corresponds to the minimum value of the energy function $E_\Omega$, the sampling problem becomes what one usually refers to as *optimization*.

To solve this problem, we employ the well-established Markov Chain Monte Carlo (MCMC) method [20]. At each iteration $n$, a proposal is generated by mutating the sequence of a chain $C_i^n \to C_i^{n+1}$ using a mutation operator $\mathcal{M}$. The total energy change is thus $\Delta E = E_\Omega(\{C\}^{n+1}) - E_\Omega(\{C\}^n)$, leading to an acceptance with the standard Metropolis criterion:

$$p_{\text{accept}} = \min\left(1, \exp\left(-\frac{\Delta E}{T}\right)\right) \tag{9}$$

where $T \geq 0$ is an effective temperature schedule controlling the probability of accepting proposals when $\Delta E > 0$. The sampling properties of MCMC guarantee that, for a large number of transitions, the system will relax to an equilibrium distribution that is the solution of the sampling problem. Moreover, if we replace $T$ with an appropriate temperature schedule $T_n$, e.g., slowly bringing temperature from some finite initial value to zero, the MCMC method becomes an optimization method able to demonstrably find the global minimum of the energy function (although possibly for an extremely large number of steps).

We summarize the algorithm as:

- **Setup.** Specify the set of States $\{S\}$ in terms of their energy terms $\epsilon_{js}$, weights $w_{js}$ and the subset of the System chains $\{C\}_S$ belonging to them.
- **Input.** Initialize the initial collection of chains $\{C\}^0$.
- **Sampling loop (MCMC).** For iterations $n = 0, \ldots, N$:
  1. Propose $\{C\}^{n+1} = \mathcal{M}(\{C\}^n)$ by mutating one or more chains.
  2. Evaluate $\Delta E = E_\Omega(\{C\}^{n+1}) - E_\Omega(\{C\}^n)$.
  (a) Obtain oracle representations $y_s$.
  (b) Compute State energies $E_S(\{C\}_S) = \sum_j w_{js} \epsilon_{js}(\{y_s\})$.
  (c) Sum over States to get System energy $E_\Omega = \sum_S E_S(\{C\}_S)$.
  3. Accept the proposal with probability $p_{\text{accept}} = \min\{1, \exp(-\Delta E/T_n)\}$; otherwise retain $\{C\}^n$.
- **Output.** Return either the State $\{C\}_S^*$ corresponding to the lowest-energy encountered (optimization) or a collection of States (sampling) once the System energy has reached its equilibrium value. In this latter case, equilibrium is obtained when the System's energy reaches a plateau, and then randomly oscillates around it.

To avoid any confusion, we stress here that within this manuscript a State (with capital S) is always used only and exclusively with this specific definition: it is a set of single-chain proteins, together with its associated set of energy terms to be evaluated given predictions on these very same chains. It is not, therefore, a reference to a conformational State of such chains. By construction, all energy terms are, in a broader mathematical sense, simply scalar potentials bounded by below. In other words, their value depends only on the current state of the system. However, these terms do not need to represent a physical energy in the thermodynamic sense – these physical energies are instead a more stringent subset of our energies. Importantly, as long as all terms are bounded scalar potentials, strong theoretical convergence results apply to Monte Carlo sampling.

## 2.2 Modular blocks

To support flexible, extensible, and composable protein design workflows, `BAGEL` is built around a set of modular, object-oriented abstractions that cleanly separate concerns such as sequence representation, mutation logic, structural prediction, and optimization/sampling. This modularity enables users to define custom protein engineering problems by composing high-level components, while also making it easy to integrate new methods and models as they emerge.

At its core, the energy is always evaluated at the `System` level, which corresponds to the total energy function $E_\Omega$. A `System` consists of multiple `State`s, each associated with its own State energy $E_S$, defined as a weighted sum over a set of `EnergyTerm`s $\epsilon_{js}$. A `State` can have one or more `Chain`s, each representing a sequence of `Residue`s, i.e. $C_i$. A `Residue` can be either mutable (i.e., allowed to mutate via a `MutationProtocol`) or immutable, depending on the design goal. Since not all `Oracle`s support all `EnergyTerm`s, and different `Oracle`s may be used to compute the same type of `EnergyTerm`, each `EnergyTerm` must be explicitly associated with a specific `Oracle`. For instance, structure-based `EnergyTerm`s require a `FoldingOracle`, while terms like `EmbeddingsSimilarityEnergy` require an `EmbeddingOracle`. Based on the total energy of the `System`, a `Minimizer` algorithm explores the sequence space using proposed perturbations from `MutationProtocol`s.

## 2.3 Oracles

An `Oracle` is any algorithm supplying data required by `EnergyTerm`s. For this initial release, we implement one `FoldingOracle` based on ESMFold and one `EmbeddingOracle` based on ESM-2, both using models from Hugging-Face's `transformers` library [21]. To support modularity and extensibility, we developed a standalone Python package, `boileroom` [19], which provides a unified API for the inference of deep learning protein models. This abstraction enables seamless swap-in/swap-out of different models within a design pipeline, making it straightforward to benchmark alternatives in terms of both predictive accuracy and computational speed. `boileroom` integrates with the serverless GPU platform Modal, enabling scalable inference either locally or on-demand, without burdening the user with hardware or dependency management.

**ESMFold** [3]. Provides 3D coordinates of all heavy atoms $\mathbf{X}$ and associated confidence metrics $\Phi$, specifically local pLDDT, pTM and PAE, which are directly consumed by structure-based `EnergyTerm`s and are described in their respective paragraphs in Sect 2.4. While ESMFold produces per-atom pLDDT scores, we default to using the $C_\alpha$-based pLDDT as a coarse proxy for local confidence. As ESMFold does not explicitly support multimer complex prediction, one has to *trick* the model by passing a single sequence, which can, however, somehow recapitulate a multimer structure. First, one can add a linker inbetween chains, usually composed of glycines [22], which adds a relative flexibility between the individual monomer chains. Second, one can introduce a gap in the positional embeddings inbetween the monomers, thus informing the model that the monomers should act almost independently, as they would effectively form a single chain, but would be far away in the primary structure (i.e., the sequence) The latter automatically ensures the backbones' of the monomers are not connected via a peptide bond. `boileroom` gives users the ability to both specify the exact sequence of the linker [23], as well as the skip in the positional encoding indices. We default to no glycine linker, and a positional encoding skip in the positional integers of 512. When using a glycine linker, the attention in the folding trunk is masked for all linker residues.

**ESM-2** [3]. Produces an informative high-dimensional embedding vector $\vec{z}$ that can be used for downstream tasks within `EnergyTerm`s. For multimers, we use the same approach as for ESMFold. We default to the 650M parameter version of the model, but leave the user the possibility to employ any of the different ESM-2 models.

## 2.4 Energy terms

Below we list the implemented `EnergyTerm`s. For clarity, we categorize them into sequence-only terms, structure-based terms, folding-metric terms, and vector-embedding terms. We define a *residue group G* as a set of residues that

contribute to an `EnergyTerm`. All residues in a group must belong to the same `Chain`, but they are not required to be contiguous in sequence. Conceptually, we adopt an N-body formalism similar to that used in molecular simulations: zero-body terms apply globally to all residues in a `State`, single-body terms act on a single residue group, and two-body terms operate on a pair of residue groups. Although this classification is not explicitly enforced in the code, it serves as a helpful abstraction when constructing and reasoning about `EnergyTerm`s. This N-body energy formulation and its modular implementation constitute a key difference compared to the original framework presented in [18], where defining certain energy terms was extremely cumbersome, especially when this definition required specifying a large number of non-contiguous regions along a single chain. More generally, their reliance on a tree graph-based representation of the system made input files increasingly difficult to construct and interpret for large systems.

### 2.4.1 Sequence terms.

**HydrophobicEnergy.** Measures the fraction of hydrophobic residues in a residue group $G$. Optionally, the magnitude of this energy can be scaled by the mean normalized solvent-accessible surface area (SASA) [24], described in detail in `SurfaceAreaEnergy` in Sect 2.4.3. In this mode, hydrophobic residues that are buried (low SASA) contribute less, while surface-exposed hydrophobes contribute more, effectively focusing the penalty on exposed hydrophobic patches. Therefore, this term can be used to penalize hydrophobicity on the surface, acting more as a structural constraint, though we keep it here for simplicity.

$$E_{\text{hydrophobic}} = \frac{N_{\text{hydrophobic}}}{N_G} \tag{10}$$

where $N_G$ is the number of residues in the residue group and $N_{\text{hydrophobic}}$ is the count of hydrophobic residues in the group. Hydrophobic residues are defined as valine, isoleucine, leucine, phenylalanine, methionine, and tryptophan, which are regarded as the most strongly hydrophobic. Users may, however incorporate mildly hydrophobic residues such as tyrosine, cysteine, alanine, and glycine in their counting or, alternatively, implement custom terms that employ a more continuous hydrophobicity measure.

**ChemicalPotentialEnergy.** Adds an energy proportional to the deviation of the number of residues from a target size, optionally raised to a power. This mimics a chemical potential term in the free energy, allowing to run an optimization with a freely floating number of residues, i.e., allowing for insertions and deletions, equivalent to sampling within the grand-canonical ensemble.

$$E_{\text{chem}} = \mu \left| N_{\text{total}} - N_{\text{target}} \right|^p \tag{11}$$

where $\mu$ is the chemical potential, $N_{\text{total}}$ is the total number of residues in the `System`, $N_{\text{target}}$ is the target number of residues, and $p$ is an exponent parameter.

### 2.4.2 Folding metric terms.

**PTMEnergy.** Penalizes low predicted Template Modeling (pTM) scores, conceived in the original AlphaFold2 paper [1]. pTM reflects the global confidence of the folding model in the predicted structure, thus it is a single scalar value over the entire `State`. The metric is based on the original TM-score [25]. This is used to achieve global structures the `FoldingOracle` is confident in.

$$E_{\text{pTM}} = -\text{pTM} \tag{12}$$

**PLDDTEnergy.** Penalizes low predicted Local Distance Difference Test (pLDDT) scores, which indicate local structural confidence. This confidence metric, originally also proposed in AlphaFold2 [1] is based on the Local Distance Difference Test [26]. The energy is the negative mean pLDDT over the residue group $G$. This term is used to promote model's confidence in a local structure.

$$E_{\text{pLDDT}} = -\frac{1}{N_G} \sum_{\alpha \in G} \text{pLDDT}_\alpha \tag{13}$$

 

where $N_G$ is the number of residues in the group, and $\text{pLDDT}_\alpha$ is the pLDDT of residue $\alpha$. To apply a global (0-body) constraint across the entire protein, the user can instead use `OverallPLDDTEnergy`, which applies the same formulation over all residues in the `System`.

**`PAEEnergy`.** Measures the uncertainty in the predicted distances between two residue groups $G_1$ and $G_2$, using the mean Predicted Alignment Error (PAE). This term encourages confident, well-defined relative positioning of residue groups, in practice, encouraging groups of residues to behave as a single rigid body, where all positions are perfectly correlated. This term is used to create protein-protein interfaces, for instance for peptide binders, and is sometimes referred to as interface PAE (iPAE), or cross PAE. It can optionally be used on a single residue group, when the pairs of residues considered are thus with the same group itself.

$$E_{\text{PAE}} = \frac{1}{N_{\text{pairs}}} \sum_{(\alpha,\beta)\in(G_1,G_2)} \frac{\text{PAE}_{\alpha\beta}}{\text{PAE}_{\text{max}}} \tag{14}$$

where $\text{PAE}_{\alpha\beta}$ is the error between residues $\alpha$ and $\beta$, $\text{PAE}_{\text{max}} \approx 30$ Å is the maximum predicted alignment error between any two residues, and $N_{\text{pairs}}$ is the total number of residue pairs considered.

### 2.4.3 Structural terms.

**`SurfaceAreaEnergy`.** Penalizes exposure of a residue group $G$ by computing its residues' mean solvent-accessible surface area (SASA), normalized by a reference maximum value. This energy encourages buried or compact structures and can be applied globally (0-body) or to specific residue groups (1-body). The SASA is calculated using the Shrake–Rupley "rolling probe" algorithm [24], as implemented in `Biotite` [27], with an adjustable probe radius.

$$E_{\text{SASA}} = \frac{1}{N_{\text{atoms},G}} \sum_{a\in G} \frac{\text{SASA}_a}{\text{SASA}_{\text{max}}} \tag{15}$$

where $\text{SASA}_a$ is the computed SASA of atom $a$, $\text{SASA}_{\text{max}}$ is a normalization constant (by default, the full surface area of a sulfur atom) in Å units, and $N_{\text{atoms},G}$ is the number of atoms in the residue group $G$ considered.

**`SeparationEnergy`.** Controls the spatial separation between two residue groups $G_1$ and $G_2$ by penalizing the Euclidean distance, in units of Å, between their backbone atoms' centroids.

$$E_{\text{sep}} = \|\mathbf{c}_1 - \mathbf{c}_2\| \tag{16}$$

where $\mathbf{c}_1$ and $\mathbf{c}_2$ are centroids of the two residue groups respectively.

**`GlobularEnergy`.** Favors globular structures by minimizing the variance of distances from backbone atoms to the centroid, promoting compact and spherical distributions. It is effectively proportional to the moment of inertia of the structure.

$$E_{\text{glob}} = \text{std}\left(\|\mathbf{x}_a - \mathbf{c}\|\right)_G \tag{17}$$

where $\mathbf{x}_a$ is the position of backbone atom $a$, $\mathbf{c}$ is the centroid of all backbone atoms in the residue group $G$, and $\text{std}(\cdot)_G$ represents the standard deviation over all backbone atoms $a$ in residue group $G$.

**`TemplateMatchEnergy`.** Drives the structure of residue group $G$ to match a provided structural template by minimizing the root mean square deviation (RMSD) between corresponding residues. RMSD is computed after optimal superposition. By default we use all the heavy atoms to obtain the alignment and the RMSD, but one can optionally choose to only use backbone atoms.

$$E_{\text{template}} = \sqrt{\frac{1}{N_{\text{atoms},G}} \sum_{\beta\in G} \sum_{a\in\beta} \|\mathbf{x}_{a,\beta} - \hat{\mathbf{x}}_{a,\beta}\|^2} \tag{18}$$

where $\mathbf{x}_{a,\beta}$ is the position of atom $a$ in residue $\beta$, $\hat{\mathbf{x}}_{a,\beta}$ is the corresponding position in the template after optimal alignment, and $N_{\text{atoms},G}$ is the total number of atoms in the group $G$. The energy can also be computed using pairwise distance matrices (distograms) instead of direct atom positions.

$$E_{\text{template}}^{\text{dist}} = \sqrt{\frac{1}{N_{\text{atoms},G}(N_{\text{atoms},G}-1)/2} \sum_{\beta \in G} \sum_{\substack{a,c \in \beta \\ a<c}} \left(\|\mathbf{x}_{a,\beta} - \mathbf{x}_{c,\beta}\| - \|\hat{\mathbf{x}}_{a,\beta} - \hat{\mathbf{x}}_{c,\beta}\|\right)^2} \quad (19)$$

where $\|\mathbf{x}_{a,\beta} - \mathbf{x}_{c,\beta}\|$ and $\|\hat{\mathbf{x}}_{a,\beta} - \hat{\mathbf{x}}_{c,\beta}\|$ are the pairwise distances between atoms $a$ and $c$ of residue $\beta$ in the current structure and template, respectively.

`SecondaryStructureEnergy.` Encourages residues in the specified group to adopt a target secondary structure (alpha-helix, beta-sheet, or coil) by penalizing deviations from the desired assignment. The secondary structure of each residue is determined using the P-SEA algorithm [28] implemented in `Biotite` [27]. The energy is defined as the fraction of residues whose assigned secondary structure does not match the target.

$$E_{\text{SS}} = \frac{1}{N_G} \sum_{\alpha \in G} \mathbb{I}\left[\text{SS}_\alpha \neq \text{SS}_{\text{target}}\right] \quad (20)$$

where $\text{SS}_\alpha$ is the assigned secondary structure of residue $\alpha$ (as a categorical label), $\text{SS}_{\text{target}}$ is the desired type (e.g., alpha-helix), $\mathbb{I}$ is the indicator function, and $N_G$ is the number of residues in the group.

`RingSymmetryEnergy.` This energy term enforces $N$-fold symmetry, where $N$ is the number of independent residue groups $G$ specified as input. Symmetry among residue groups $G$ is encouraged by minimizing the variance in distances between the groups' centroids - either for all pairs of $G$ or only their direct neighbors. We define direct neighbours as consecutive groups $G$ specified in the residue group list in the input. The first and last element of the list are also considered direct neighbors.

$$E_{\text{sym}} = \text{std}\left(d_{G_\alpha G_\beta}\right)_G \quad (21)$$

where $d_{G_\alpha G_\beta} = \|\mathbf{c}_{G_\alpha} - \mathbf{c}_{G_\beta}\|$ is the distance between centroids of the pairs of groups $G_\alpha$ and $G_\beta$, computed by backbone atoms only, and $\text{std}(\cdot)_G$ represents the standard deviation over all the group pairs considered. Referring back to the N-body notation, notice that this term is an arbitrary N-body term, given its flexibility to include an arbitrary amount of groups $G$, achieving an arbitrary $N$-fold symmetry.

### 2.4.4 Embedding terms.

`EmbeddingsSimilarityEnergy.` Measures the cosine similarity between learned embeddings of the current sequence and a group of reference embeddings, encouraging functional or structural similarity at the embedding level.

$$E_{\text{embedding}} = 1 - \frac{1}{N_G} \sum_{\alpha \in G} \cos \theta_\alpha \quad (22)$$

where $\theta_\alpha$ is the angle between the high-dimensional embeddings of residue $\alpha$ in the current sequence and reference sequences respectively as $e_\alpha$ and $e_\alpha^{\text{ref}}$. $\cos \theta_\alpha = e_\alpha \cdot e_\alpha^{\text{ref}}$ is then the cosine similarity and $N_G$ is the number of residues in the group. Note that this energy term is defined so that its minimum is exactly zero when all embeddings are equal to the reference values.

## 2.5 Mutation protocols

`MutationProtocol`s define how to perturb sequences during optimization. Each protocol implements a strategy for proposing mutations to the sequence, such as point substitutions, insertions, or deletions. These proposals form the basis

of the sampling procedure used by the `Minimizer`. All protocols operate on `Chain`s within a `System`, ensuring consistency across all `State`s sharing the same `Chain` object. In other words, when a chain is mutated, all States containing said chain are automatically updated.

**`Canonical.`** Implements fixed-length point-mutation moves. At each optimization step, we:

(i) pick a `Chain` with probability proportional to its number of mutable `Residue`s;
(ii) draw $n_{mut}$ `Residue`s uniformly from that `Chain`;
(iii) resample their amino acid identity from a user-defined categorical distribution $p_{mut}$, which specifies the mutation probability for each amino acid type.

We default to $p_{mut}$ that avoids mutating to cysteines to prevent downstream effects, such as disulfide bond formation.

**`GrandCanonical.`** Extends the canonical scheme with length-changing moves, thereby sampling a grand-canonical ensemble in which sequence length is conjugate to an effective chemical potential $\mu$, described in the `ChemicalEnergyTerm` above. For each of the $n_{mut}$ attempted moves, we:

(i) select a `Chain`;
1. draw a move type $m_{move} \in \{\text{substitution}, \text{addition}, \text{removal}\}$ with user-defined probabilities $p_{type}$;
(ii) execute one of the following:
(iii) substitution: identical to `Canonical`.
   - addition: pick an insertion index $\alpha \in \{0, \dots, L\}$ from $L + 1$ indices, sample an amino acid type with $p_{mut}$, and insert a new `Residue` at index $\alpha$.
   - removal: choose a mutable `Residue` at index $\alpha$ uniformly and delete it, provided the `Chain` retains length greater than one (i.e., is not deleted completely from the `State`).

Upon addition of a new `Residue`, we determine whether it should *inherit* the influence of `EnergyTerm`s from its neighbouring residues. If both neighbours (or one, at `Chain` boundaries) are associated with the same `EnergyTerm`, the new residue inherits it. If the neighbours differ in their `EnergyTerm` assignments, the new residue inherits from either neighbour with equal probability. As a result, each `EnergyTerm` must specify whether it is *inheritable*. Most terms are inheritable by default. However, some, such as `TemplateMatchEnergy`, are not since including newly inserted `Residue`s in a comparison with a fixed-length reference structure would be ill-defined. Note that `GrandCanonical` can be used without a `ChemicalEnergyTerm`, in which case it implicitly runs with a zero chemical potential.

## 2.6 Minimizers

`Minimizer`s define the sampling/optimization strategy used to explore sequence space (and minimize the energy function) defined over a `System`. Each `Minimizer` implements a protocol for proposing and accepting mutations over a series of steps. Below, we describe the currently available `Minimizer`s.

**`MonteCarloMinimizer.`** Implements a general Monte Carlo sampling protocol with pluggable acceptance criteria. By default, it uses the standard Metropolis criterion $p_{n \to n+1} = \min\left(1, \exp\left(-\frac{\Delta E}{T}\right)\right)$. Users have full control over the temperature schedule, which may be constant, annealing, or arbitrarily defined. While called a *minimizer*, this class naturally generalizes to sampling applications, depending on the choice of temperature schedule, shown in the enzyme variant generation example in Sect 3.4. To encourage retention of promising candidates, the best `System` found so far can be periodically reinstated as the current step in the optimization by specifying an interval $n_{\text{best system}}$.

**`SimulatedAnnealing.`** A special case of `MonteCarloMinimizer` that employs a linearly decreasing temperature schedule from an initial high temperature to a final low temperature.

 

**SimulatedTempering.** A specialized variant of the `MonteCarloMinimizer`, this protocol employs a cyclical temperature schedule alternating between fixed low and high temperatures, $T_{low}$ and $T_{high}$, across $n_{cycles}$ cycles. Each cycle comprises $n_{low,steps}$ steps at low temperature followed by $n_{high,steps}$ at high temperature, always beginning in the low-temperature regime. To encourage retention of promising candidates, we set $n_{best\,system}$ to the total number of steps in one full cycle. This ensures the preservation step always occurs at the end of the high-temperature phase. Anecdotally, this strategy has led to improved convergence toward viable protein candidates.

## 2.7 Analyzers

We provide an abstract class `Analyzer`, which serves as the starting point to develop reliable data analysis tools for the output files. We currently support outputting the current `System` and the best `System` found so far, including their FASTA sequences, CIF structure files, and ESMFold-related metrics like pLDDT and PAE matrices, stored as a `NumPy` [29] text file. The exact outputs are as follows.

`config.csv`: A CSV file listing each `State`, `EnergyTerm` name, and its numerical weight. In practice, this file contains all the data to reproduce (at least in a statistical sense), any given experiment, thereby helping reproducibility. Nevertheless, we recommend saving the original Python script to ensure reproducibility, until a more reliable solution - such as standardized `json` config files - is implemented.

`current/energies.csv` and `best/energies.csv`: For both the running ("current") and optimal ("best") `System`s, a CSV file where each row has individual energy contributions of each `State` at a particular step (e.g., `stateA:HydrophobicEnergy`).

`current/state.fasta` and `best/state.fasta`: FASTA files for each `State`, where each entry - row - represents a specific minimization step, and multiple `Chain`s are concatenated and separated with "`:`".

`current/structures/` and `best/structures/`: Directories containing `State`'s structure as a CIF file per minimization step, named `<State>_<Oracle>_step.cif`, together with any `Oracle`-specific outputs (e.g., PAE or pLDDT arrays saved as `.pae` and `.plddt` files in a `NumPy` text format respectively).

`optimization.log`: A CSV file with one row per MC step, containing information from the `Minimizer`, including the temperature, and whether the step was accepted or rejected.

## 3 Results

To illustrate the utility of `BAGEL`, we present design tasks that serve as reference workflows and starting points for new applications. These examples span common protein engineering use cases and demonstrate various framework features. All are reproducible, modifiable, and provided as ready-to-run templates in the GitHub repository. Optimization details, along with an input file, are included in the S1 Text. While not experimentally validated, these designs are valid under the assumption that ESMFold and ESM-2 yield accurate predictions. Specifically, one can monitor the expected success of these designs by referring to the evolution of the confidence metrics, e.g., pLDDT, iPAE, or pTM, during optimization. As extensively shown (see [30,31]) by comparison between predictions and experiments, low values of the predicted confidence metrics strongly correlate to lower RMSD values between the predicted and experimentally determined structure, at least in the case of single-chain predictions for natural proteins. While a large-scale assessment of the translational accuracy of these metrics remains to be done for the case of designed proteins, the current available data show that high confidence metrics are indeed correlated to more successful designs, especially for the relevant case of protein binders. Nevertheless, to obtain an orthogonal validation of the stability of the designed sequences, as in [32,33], we perform molecular dynamics trajectories, showing the structures are at least stable on the simulated timescales (details of all simulations are provided in S3 Text).

### 3.1 Simple peptide binder

A primary application of BAGEL is designing small proteins that can bind a protein target of interest. Fig 2 shows binders to three clinically relevant targets: carbonic anhydrase IV (CA4, UniProt P22748), a protein associated to hypertension and cardiovascular disease [34], epidermal growth-factor receptor (EGFR, UniProt P00533), an important target for cancer therapeutics [35], and the dust-mite allergen Der f 7 (DERF7, UniProt Q26456), studied for the neutralisation and alleviation of allergic reactions [36]. All were designed to minimize SeparationEnergy and PAEEnergy (interface) in respect to the binder and the hotspot on the target. Taken together, these examples highlight BAGEL's ability to explore helical, beta-rich, and coil-dominated solutions from the same generic energy function.

### 3.2 Targeting intrinsically disordered epitopes

In our earlier work [37], we showed how BAGEL's energy functions can lead to peptide binders design that induce order in an otherwise intrinsically disordered region (IDR) of a protein. We designed several such peptides for four clinically relevant targets - $\alpha$-synuclein (ASYN, UniProt P37840), T-cell co-receptor CD28 (UniProt P10747), tumour suppressor p53 (P53, UniProt P04637) and Small Ubiquitin-like Modifier 1 (SUMO, UniProt P63165). Traditional drug discovery approaches require docking of molecules into rigid pockets, thus having the ability to bind disordered regions opens up plethora of applications for therapeutic modulation in disease [38], ranging from neurodegenerative diseases (ASYN) [39] through immune system regulation (CD28) [40] to cancer (P53, SUMO) [41]. Binding to disordered epitopes is achieved by enforcing high pLDDT values on the IDR with PLDDTEnergy, and optionally also promoting secondary structure with SecondaryStructureEnergy. In our original work, we also validated these designs and their binding affinities with free energy calculations. Fig 3 illustrates the key effect: the presence of the binder sharply increases the pLDDT confidence - either across the full target (ASYN), or locally at the targeted IDRs (CD28, P53, SUMO) - highlighting BAGEL's ability to stabilize otherwise disordered segments.

### 3.3 Multi-state selective peptide binder

A key strength of BAGEL is its ability to optimize a single binder sequence against multiple targets simultaneously. Here, we demonstrate a two-State design in which the binder must engage the mouse Zif268 zinc-finger domain (EGR1, UniProt P08046), while avoiding binding to the closely related human zinc-finger protein ZNF593 (ZNF, UniProt O00488). Species-selective binders like this could be useful in xenograft transplant settings, where tissue-specific or species-selective gene knock-out is desired [42,43]: for example, when humanizing the mouse immune system. We achieve this selective binding by minimizing the interface PAEEnergy for the mouse target (encouraging interaction), while simultaneously maximizing the same metric for the human off-target (discouraging binding). The latter is implemented using a negative weight $w_{js}$. Fig 4 summarizes the results. Multi-State optimization in BAGEL can be extended beyond this example - for instance, by designing cross-species (also known as cross-reactive) binders that target both mouse and human variants simultaneously. This flexibility is critical for therapeutic development, where candidates must succeed in both animal models and human trials. BAGEL's modular framework supports similar multi-functional design problems with ease.

### 3.4 Enzyme variants with conserved active site

We previously introduced a generative algorithm to design enzyme variants while preserving local function in [44], purely relying on sequence embeddings and without any structural information. Given the modular versatility of BAGEL, this method is now also implemented in this package. In the example, we demonstrate this with oxidoreductase (UniProt P0AEG4), where the active site (Cys30 and Cys33) is held immutable in sequence, while the remainder of the protein is allowed to mutate freely. To guide sampling, we use ESM-2 and EmbeddingsSimilarityEnergy, encouraging similarity to the reference protein in embedding space in the active site, while permitting extensive sequence drift elsewhere.

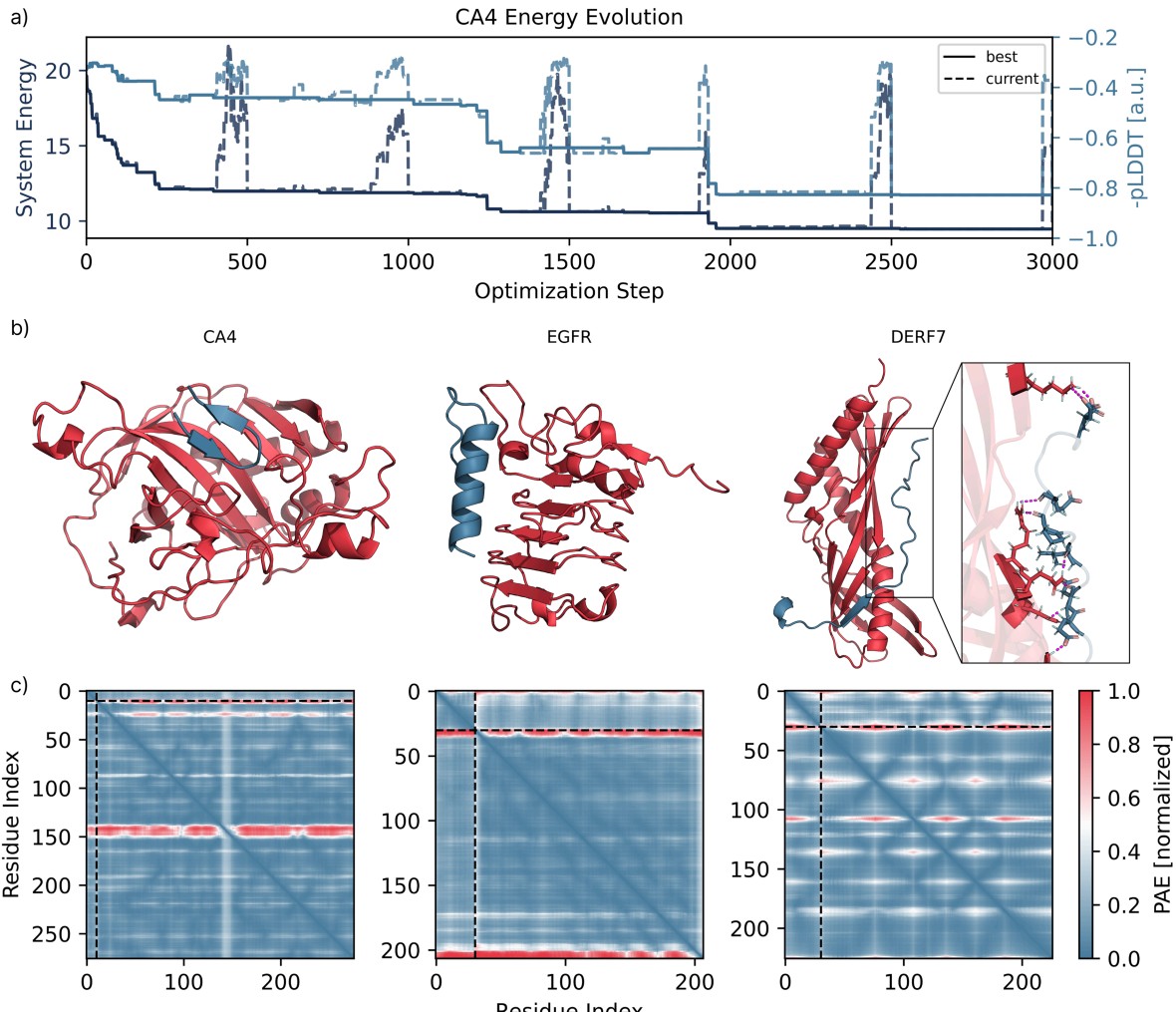

**Fig 2**. **Peptide binder design for three protein targets using `BAGEL`. a)** System energy (dark, left axis) and binder –pLDDT (light, right axis) traces during CA4 binder optimization. The optimal sequence emerged within the first ~2,000 of 20,000 steps. Increases in energy and –pLDDT correspond to the high-temperature regime. Solid lines show the best System found; dashed lines, the current System. **b)** Final binder-target complexes for carbonic anhydrase IV (CA4), epidermal growth factor receptor (EGFR), and dust-mite allergen Der f 7 (DERF7). Binders in blue, targets in red. CA4 minimization yielded beta-sheet-rich binders despite no secondary structure restraints. For EGFR, we model only Cys329–Val506, which includes a known epitope. For DERF7, no hotspot constraints were used; minimization was over the entire target. Though the DERF7 binder appears loosely structured due to its coil-like geometry, equilibrated structures (10 ns of molecular dynamics at 310K) show plausible interfaces mediated by hydrogen bonds (magenta dashed lines, close-up). The binder wraps around the target as minimizing `SeparationEnergy` without a hotspot often collapses centroids - illustrating a potential failure mode that may require more careful tuning of energy terms and weights. **c)** PAE matrices for each binder-target pair, normalized to 1. Black dashed lines delineate binder (upper-left blocks) from target. All interfaces show low iPAE values: 0.242, 0.255, and 0.235, respectively.

In contrast to the previous applications and converging to a single optimum, we use the `MonteCarloMinimizer` to collect a diverse ensemble of sequences that satisfy the design constraints at constant temperature. Fig 5 summarizes the results of this simulation. Despite large differences in the sequence and mutable structure, the catalytic site is preserved both geometrically and chemically, demonstrating the ability of embedding-guided sampling to produce viable functional diversity. As described in the original work [44], this *in silico* sampling can effectively *seed* rounds of directed evolution in the lab, allowing experimentalists to focus experimental screening on useful mutations, and avoiding deleterious mutations that would destabilize or unfold the active site altogether.

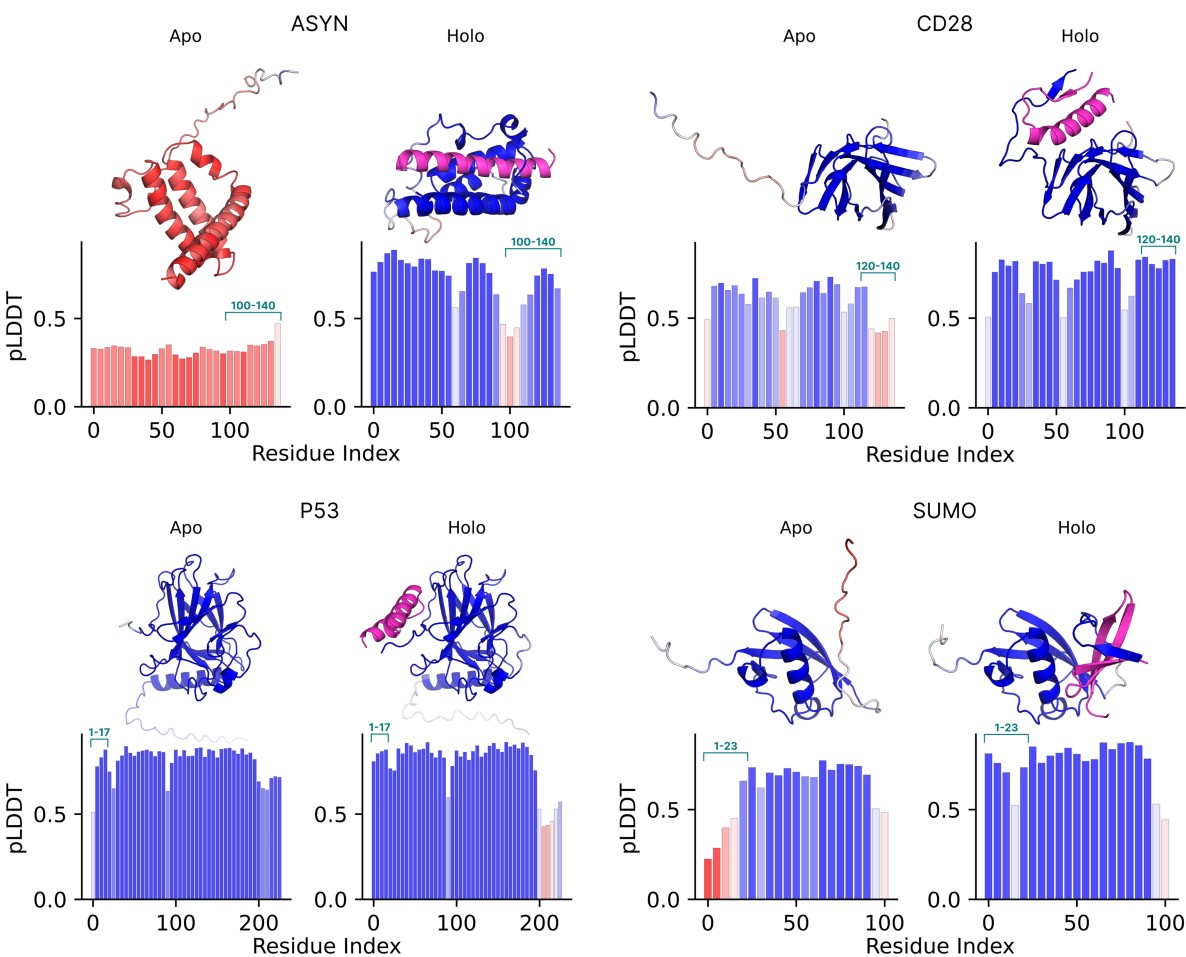

**Fig 3**. **Targeting of intrinsically disordered epitopes with designed binders using `BAGEL`.** For each of the four targets in the quadrants - $\alpha$-synuclein (ASYN), CD28, p53 (P53) and SUMO-1 (SUMO) - the left sub-panel shows the predicted *apo* structure, while the right sub-panel (*holo*) includes the magenta peptide binder. Backbones are colored by per-residue pLDDT confidence (red → white→ blue corresponds to 0.0 to 1.0 transition). The bar plots beneath each structure display the same pLDDT values averaged over consecutive 5-residue windows, illustrating the local gain in structural confidence upon the presence of the binder targeting the IDR epitopes highlighted in cyan. Note that the horizontal axis is 0-indexed, while the epitope residue numbers in cyan are 1-indexed. ASYN shows a broad increase in pLDDT across the whole target, while P53 shows a significant reduction in pLDDT on the opposite end of the sequence (beyond residue 200) from the targeted epitope. CD28 and SUMO show the expected behavior, with increased pLDDT localized to the epitope region. Figure adapted from [37].

## 4 Availability and future directions

We built `BAGEL` with the clear goal not only to withstand the test of time, but to actually mature and improve as time passes, as more accurate and faster algorithms evaluating properties of proteins become available, or with advancements in computing hardware. Nevertheless, the molecular community's skepticism – summarized in the aphorism "garbage in, garbage out" – remains valid, and `BAGEL`'s prediction can only be as good as the currently available tools allow. In this regard, we recognize several current limitations: some internal - directly stemming from our choices and omissions for this initial release; and some external - dependent on the accuracy of current models, especially when considering truly de novo protein sequences and multimers, and on their ability to provide accurate predictions, e.g., of the binding interactions

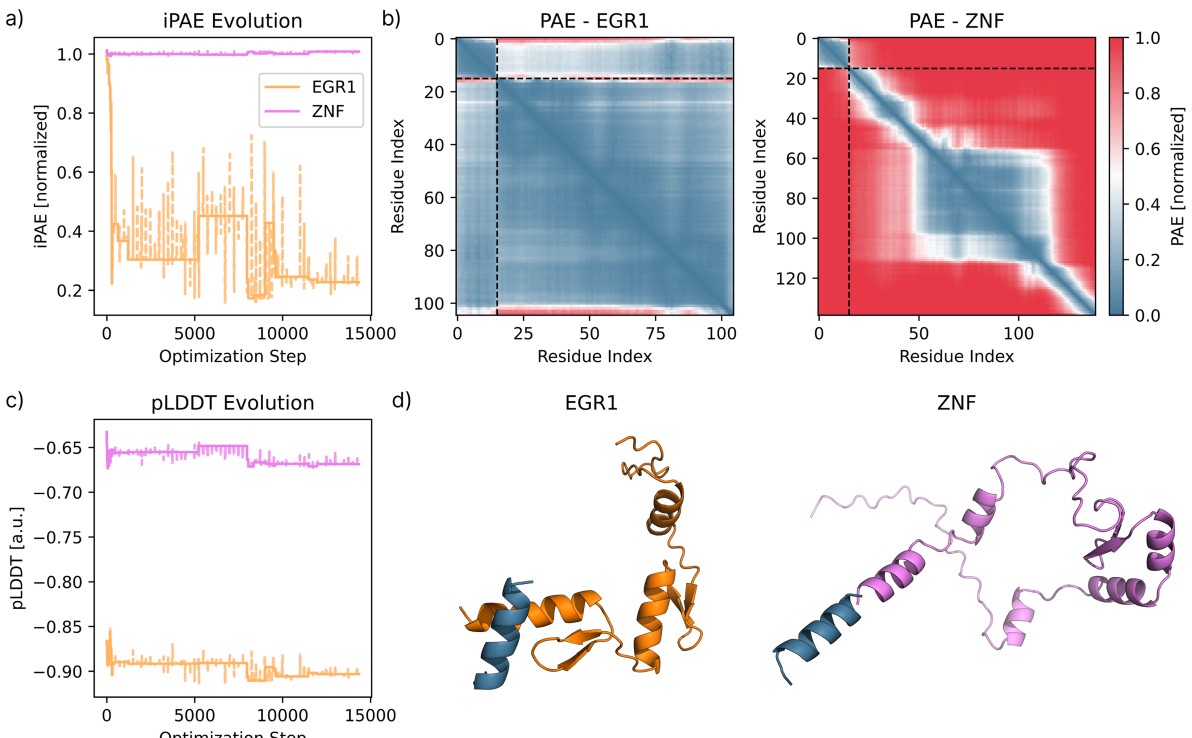

**Fig 4.** **Selective binder design against targets, avoiding off-targets using `BAGEL`. a)** Evolution of the mean interface (iPAE) for the binding (orange) and non-binding (violet) States between the binder and the target. Solid lines are the best energies, and dashed lines are the current energies from the MC trajectory. **b)** Final PAE matrix for the binding (EGR1) and non-binding (ZNF) States; black dashed lines separate binder (upper-left block) from target residues. The final mean iPAE of the binding and non-binding States are 0.23 and 1.0 respectively (normalized units), observable in the PAE matrices. **c)** Evolution from the MC optimization of the binder's mean pLDDT in each State, colors are the same as in a). **d)** Structure of the binding and non-binding complex (dark blue = binder, orange = target, violet = target to avoid). From further visual inspection, only the binding pair shows a well-packed contact surface, while the non-binding State renders an implausible, sterically clashing structure. The pLDDT of the ZNF binder is relative low (0.66 as shown in a)), while the iPAE is relatively high (shown in b)). All of these hint at a low chance of the binder interacting with the off-target ZNF.

between different proteins. The ongoing development of `BAGEL` is envisioned to address these limitations to ensure continuous relevance and adaptability in the field of protein engineering. We note, however, that even with continued development, BAGEL does not constitute a plug-and-play package to any protein design problem, and should be used with a wider set of computationally available tools, such as PESTO [45] to find binding sites, or sequence inpainting models [11–13] to increase protein candidate diversity. In the long-term, on the other hand, all these could in principle be employed within the BAGEL framework, as a part of a broader protein design optimization campaign.

**Improved `Oracle` Models.** Future versions will integrate additional state-of-the-art `FoldingOracle`s - a non-exhaustive list to benchmark includes RGN2 [46], AlphaFold2/3 [1,47], Chai-1 [48], and Boltz-1/2 [49,50]. In this regard, unifying the inference access with `boileroom` will enable direct comparison of the models' accuracy and computational speed. Given the sequential nature of MC, computationally faster models such as MiniFold [51] also need to be considered, even at the expense of marginal inaccuracy. Moreover, one could only calculate the change in mutated embeddings instead of recomputing their value from scratch (an interpolation of a Car-Parrinello-like approach to embeddings [52]), potentially leading to increased inference speed. We also point out that different models excel in different scenarios: for example, RGN2 has been shown to outperform other models for orphan proteins structure prediction [46]. Therefore, one approach is to employ an ensemble of `FoldingOracle`s with a weighted consensus; although it remains unclear whether such consensus improves design outcomes [53]. Once additional `FoldingOracle`s are implemented inside

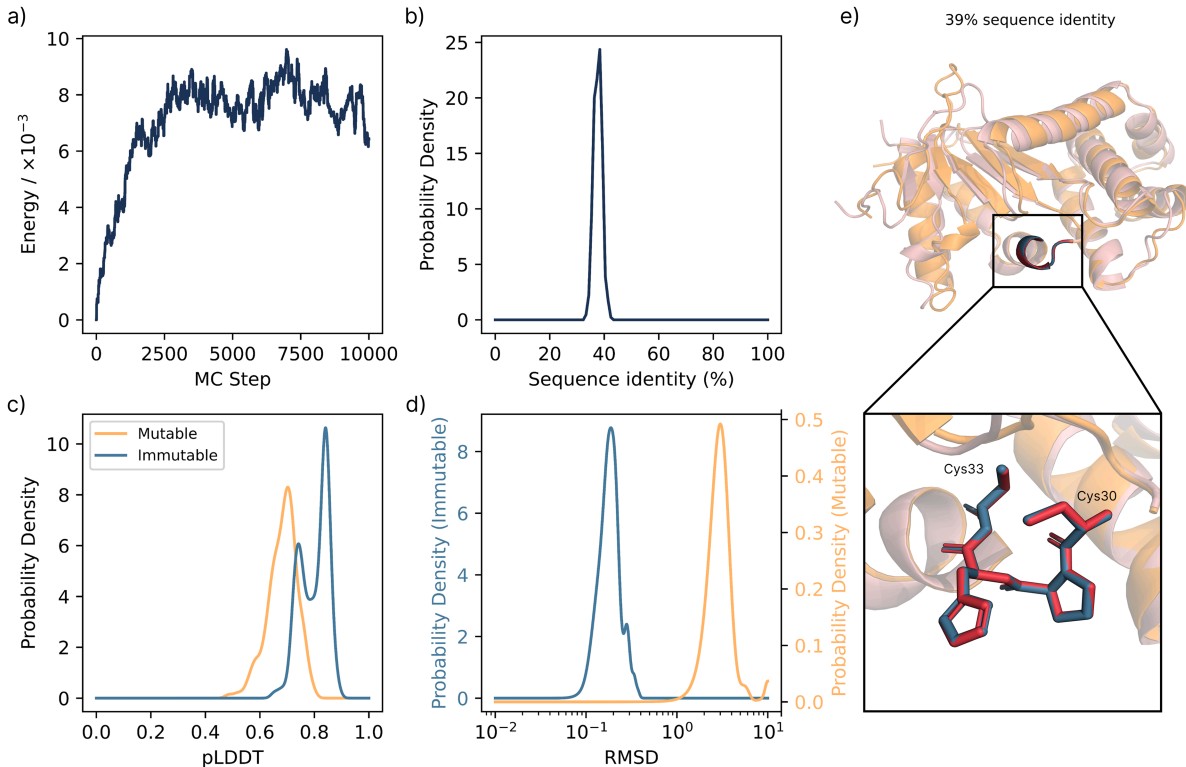

**Fig 5**. **Enzyme (oxidoreductase) variant generation with BAGEL. a)** Energy evolution of the MC trajectory to sample new enzyme variants. We define the first 3,000 steps as the *transient* region and analyze the subsequent *equilibrium* sampling from steps 3,000 to 10,000, collecting 1,286 unique variants as a result. **b)** Probability density of the sequence identity to the original sequence. **c)** Probability density of pLDDT values comparing the residues within the mutable and immutable groups. Immutable residues, i.e., the active site, show significantly higher pLDDT values. **d)** Probability density of RMSD to the original sequence. All RMSD values are computed using folded structures with ESMFold, including the original sequence. Mutable residues' RMSD is computed on the backbone atoms, immutable residues' RMSD is computed on all the heavy atoms of the active site, i.e., Cys-30 and Cys-33. Structures are superimposed to minimize the RMSD of the active site. Despite the relatively low sequence identity, the RMSD of the active site remains low, and the pLDDT sufficiently high. **e)** Example of a variant (blue = original, red = variant), including a close-up of residues Cys-30 to Cys-33. For this variant, the RMSD of the immutable and mutable regions are 0.11 Å and 2.7 Å respectively.

BAGEL, such an approach can be readily implemented within the current codebase. Regarding EmbeddingOracles, we plan to explore models including AMPLIFY [54] and ProtGen [55], as well as protein language models trained explicitly to provide improved descriptions of protein-protein interactions such as the recently released MINT [56]. In general, the modular nature of BAGEL will help us leverage the continuous improvements of models and hardware in the upcoming months and years; and given that we do not rely on gradient-based optimization, the implementation is more flexible to swap out the underlying models, enabling rapid and convenient experimentation with different models.

**Expanded Oracle Suite.** The modular design philosophy also motivates us to introduce new types of oracles, helping to provide more physically accurate information on interactions and dynamics. For instance, the promise of emerging sequence-to-ensemble predictors, such as BioEmu [57], AlphaFlow [58] or CryoBoltz [59], could be used to compute EnergyTerms over ensembles as opposed to static structures, with the goal of better capturing the *true* thermodynamic behavior of biomolecules. A more physically-grounded description will come from introducing molecular dynamics (MD)-based Oracles. Simple energy minimization with a physical force field, such as AMBER [60], would filter out unstable, or sterically clashing structures. Furthermore, a short MD simulation measuring the RMSD between the initial and equilibrated structures could be used to estimate the stability of the complex. Finally, the most computationally demanding

approach would involve techniques such as free energy perturbation [61] to provide the most correlated relative binding affinity prediction as an `EnergyTerm` to guide the MC trajectory in search of the optimal design.

**Advanced `Minimizer` Algorithms.** Although currently we only implemented different forms of Monte Carlo sampling with point mutations to explore the energy landscape, see Sect 2.1, arbitrary sampling/optimization algorithms can be employed. In the near future, we plan to benchmark genetic algorithms (GAs) and carefully evaluate their potential to improve sequence space exploration efficiently. We expect that this endeavour will require careful consideration of the crossover mutation algorithms used, especially for shorter sequences. Moreover, we plan to leverage the scalability of `boileroom` and implement parallel tempering, hence truly taking advantage of parallelization, which may help alleviate the inherent limitations of sequential, single-replica MC. Based on benchmarking the accuracy and speed of different `FoldingOracle`s, a low-hanging fruit that could also be rapidly implemented within the existing codebase is a multi-resolution sampling scheme [62,63]. First, a computationally-cheaper model, such as RGN2 or MiniFold, is initially used to generate a long Markov Chain of $n$ steps to move further and more rapidly in the energy landscape. Second, a more computationally-expensive (and, in principle, more accurate) model, such as AlphaFold3, is used to accept or reject the whole chain. This would be especially useful for systems with very long decorrelation times, when many potential iterations of the MC algorithm are wastefully spent sampling very similar candidates corresponding to the same basin of the energy landscape.

**New `EnergyTerm`s.** Solving different design problems requires the use of `EnergyTerm`s to drive the system in the correct region of sequence space. While in the current implementation of `BAGEL` we implemented useful terms to tackle various classical protein design tasks such as motif templating and binder design, there is still much work to be done. Two different but complementary directions are the introduction of terms that a) quantitively improve desired correlations for already implemented design objectives, or b) qualitatively widen the range of potential design tasks. In the first group, the highest-priority advancement includes implementation of new `EnergyTerm`s that more directly correlate with the interaction between two groups of residues, possibly calculated using `Oracle`s specialized in reproducing different aspects of protein-protein interactions, such as the previously discussed MINT [56], or PESTO [45]. In the second group, the choice is only limited by our creativity and the design problem we want to solve. For the next iteration of `BAGEL`, we plan to implement terms that can be used to enforce the positioning of specific amino acids in user-defined positions, e.g., to create addressable protein scaffolds. Combining such terms with excluded volume terms that create void regions, for example, could be used to create pockets to co-localise drugs or catalysts in a specific protein patch, which could be useful to design enzymes for pro-drugs activation [64]. Similarly, `EnergyTerm`s that measure the propensity for a given protein surface to interact with other class of molecules, such as nucleic acids, could be used to design transcription factors with known binding patches. In this endeavour, we make an open call to the community to contribute new `EnergyTerm`s relevant to their research interests, thus increasing the breadth of problems addressable with `BAGEL`.

**Choice of `EnergyTerm`s.** A current limitation of `BAGEL` lies in the need for users to carefully select both the `EnergyTerm`s and their relative weights. While this flexibility enables highly customizable design objectives, it also places the burden of defining a meaningful optimization target on the user. In practice, success often depends on choosing a metric – or an associated energy term – that best captures the desired physical or functional property. For example, when the design objective is to obtain a stable structure, terms such as pLDDT, pTM, or iPAE can serve as practical surrogates for experimental observables like RMSD or binding affinity [31,65,66]. Nevertheless, there remains to be a system-independent confidence metric coming out of these models that would serve as a one-size-fits-all proxy for experimental success [53,67–69]. Model confidence metrics remain informative but imperfect proxies for real-world biophysical accuracy. Consequently, a key future direction for `BAGEL` will be to incorporate adaptive schemes that automatically identify and weight the most predictive metrics for a given design class – potentially through Bayesian optimization, reinforcement learning, or large language model-based agents – thus improving both user experience and design reliability.

**Future of `boileroom`.** Managing dependencies across diverse deep learning models often results in complex and fragile dependency graphs. We introduced `boileroom` to alleviate the dependency management, running each model

in an isolated, containerized environment on Modal. Nevertheless, this introduces internet connection requirements and network latency, which we hope to tackle by providing better support for local execution of these models. We can achieve this by employing similar logic to Modal, where containers are spun up through conda, Docker or Apptainer, and thus run these models in isolated environments. We are also considering integrating with NVIDIA's BioNemo and NIM frameworks, which are, however, still in their infancy in terms of the breadth of bio-related models provided. Lastly, `boileroom`'s API should be unified, abstracting a consistent `Protein` class to ensure consistent and predictable outputs from the interface.

**Biomolecular modality.** Long-term additions might include the introduction of non-canonical amino acids, e.g., with RareFold [70], or nucleotides, if models supporting nucleic acids (NAs) - such as Boltz-2 and AlphaFold3 - reach a sufficiently translatable accuracy in predicting protein-NA interactions. Non-canonical-based peptides have been growing in popularity thanks to their stability and bioavailability [71], while targeting nucleic acids could lead to novel gene-editing [72] or cell reprogramming tools [73]. This would require a more detailed refactoring of the `Chain` and `Residue` logic, however, the overall codebase's philosophy and components would stay the same. At the same time, we highlight that `FoldingOracle`s or `EmbeddingOracle`s trained for systems made exclusively of DNA or RNA [74] could be directly implemented and used to design nucleotide-based structures within `BAGEL` with very minimal changes. Although they would probably require implementation of *ad hoc* `EnergyTerm`s, such an endeavour would immediately expand `BAGEL`s capability to design a new class of systems.

**Expert in the loop.** Eventually, whether or not a design will work in practice depends on two important aspects: the quality and accuracy of the Oracles used, and whether the design problem is well-described by the loss function provided by the user. Just to make an example, if a functional property of a protein is strongly dependent on its structural fluctuations, an Oracle able to evaluate these effects should be used. Failure to use the right Oracle, or failure to account for these effects in the loss function, will necessarily result in an experimentally unsuccessful design. In this regard, the continuous improvements of the community around different models, i.e., Oracles, as well as better understanding of their use and limitations, will allow users to use BAGEL better. In other words, while BAGEL allows to easily translate any design problem into a loss function, and quickly test the results of its minimization, it does not replace expert knowledge, which should be used as much as possible to pre-condition the design problem to help finding the correct solution.

**Community engagement and contribution.** We aim to cultivate a vibrant community around `BAGEL` by encouraging contributions via GitHub pull requests. Specifically, we welcome implementations of new `Oracles` into `boileroom`, new `Minimizer` algorithms or adaptations of existing ones, and detailed benchmarking studies using the suite of available components. Our principal ambition ultimately remains grounded in experimental validation. We thus aim to rigorously test the practical efficacy of `BAGEL`-designed proteins in laboratory conditions. This will help establish `BAGEL` not only as a computational tool, but as a reliable platform for practical biological innovation. Accordingly, we extend an open call to experimentalists seeking *in silico* designed peptides: the provided templates on GitHub may be readily adapted for specific applications, and we welcome direct contact to discuss potential collaborations.

## Acknowledgments

We thank Shanil Panara, Dr Daniele Visco, Arnav Cheruku and Harsh Agrawal for fruitful conversations discussing the implementation of the codebase and design pipeline. We acknowledge computational resources and support provided by the Imperial College Research Computing Service (http://doi.org/10.14469/hpc/2232).

## Supporting information

**S1 Text. Experimental details and parameters for all protein design applications.** This file contains information on all the energy terms, weights, and optimization parameters used for the simple peptide binder designs (CA4, EGFR, DERF7), targeting intrinsically disordered epitopes (ASYN, CD28, P53, SUMO1), multi-State selective binder design, and enzyme variant generation.
(PDF)

**S2 Text. An example Python input file for the CA4 binder design.**
(TXT)

**S3 Text. Details of molecular dynamics simulations for simple peptide binder design, selective binder design, and enzyme variant generation applications.**
(PDF)

**S1 Fig. Computational scaling of ESM-2 and ESMFold `Oracles`.** Time taken to infer the embeddings or structure for ESM-2 and ESMFold respectively using various GPUs. The sequence lengths used were [16, 32, 64, 128, 256, 512, 1024] amino acids.
(TIF)

**S2 Fig. Molecular dynamics simulation of the simple peptide binder design for CA4, EGFR, and DERF7.** For all three complexes, the binders remain in contact with their targets for hundreds of nanoseconds, supporting the notion of stable binding. Contacts are defined as any $C_\alpha$ atom pairs with a distance less than 8.0 Å.
(TIF)

**S3 Fig. Molecular dynamics simulation of the selective binder design.** EGR1 is the target of interest, while ZNF is the off-target. In the implicit solvent simulation, the binder forms contacts with both the target and the off-target throughout the simulation. Nevertheless, in the explicit solvent calculations, the binder loses contact with the off-target. Contacts were defined as any $C_\alpha$ atom pairs with a distance less than 8.0 Å. Dashed lines show the average of the trajectory.
(TIF)

**S4 Fig. Thermal fluctuations of the variants (DSBA-X) and wild type (DSBA) in the active site of the enzyme.** Thermal fluctuations across the four key residues (Cys30–Cys33). In implicit solvent, the designed variants (DSBA-X) show fluctuations comparable to the wild type (DSBA). However, in explicit solvent, all variants exhibit higher RMSF values than the wild type, indicating increased flexibility of the active site relative to the enzyme frame of reference. The Root Mean Square Fluctuations (RMSF) are computed using all the heavy atoms of the four key residues (Cys30–Cys33). The frames of the trajectory were aligned on the $C_\alpha$ atoms of the entire enzyme. All systems were simulated for 650 ns.
(TIF)

**S5 Fig. Structural conservation of the active site in the variants (DSBA-X) and wild type (DSBA) of the enzyme.** Strucutral deviation of the designed variants and wild type to the wild type ESMFold-derived structure. The implicit solvent show strong agreement, while the explicit solvent shows notable differences. The explicit solvent distributions could be used as a filtering metric for the designed variants. The Root Mean Square Deviation (RMSD)s are computed using all the heavy atoms of the four key residues (Cys30–Cys33). The frames of the trajectory were superimposed to the wild type structure using heavy atoms of the key residues. The distributions are shown as kernel density estimations. All systems were simulated for 650 ns.
(TIF)

## Author contributions

**Conceptualization:** Stefano Angioletti-Uberti.

**Data curation:** Jakub Lála.

**Formal analysis:** Jakub Lála, Stefano Angioletti-Uberti.

**Funding acquisition:** Jakub Lála, Stefano Angioletti-Uberti.

**Investigation:** Jakub Lála, Ayham Al-Saffar, Stefano Angioletti-Uberti.

**Methodology:** Jakub Lála, Ayham Al-Saffar, Stefano Angioletti-Uberti.

**Project administration:** Stefano Angioletti-Uberti.

**Resources:** Jakub Lála, Stefano Angioletti-Uberti.

**Software:** Jakub Lála, Ayham Al-Saffar, Stefano Angioletti-Uberti.

**Supervision:** Stefano Angioletti-Uberti.

**Validation:** Jakub Lála, Stefano Angioletti-Uberti.

**Visualization:** Jakub Lála.

**Writing – original draft:** Jakub Lála.

**Writing – review & editing:** Jakub Lála, Stefano Angioletti-Uberti.

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
