## [Decision Letter · Decision Letter 0]

10 Oct 2025

PCOMPBIOL-D-25-01443

BAGEL: Protein Engineering via Exploration of an Energy Landscape

PLOS Computational Biology

Dear Dr. Angioletti-Uberti,

Thank you for submitting your manuscript to PLOS Computational Biology. After careful consideration, we feel that it has merit but does not fully meet PLOS Computational Biology's publication criteria as it currently stands. Therefore, we invite you to submit a revised version of the manuscript that addresses the points raised during the review process.

Please submit your revised manuscript within 60 days Dec 10 2025 11:59PM. If you will need more time than this to complete your revisions, please reply to this message or contact the journal office at ploscompbiol@plos.org. Please include the following items when submitting your revised manuscript:

We look forward to receiving your revised manuscript.

Kind regards,

Amar Singh

Academic Editor

PLOS Computational Biology

Arne Elofsson

Section Editor

PLOS Computational Biology

**Additional Editor Comments:**

Thank you for your patience during the review of your manuscript. The process has taken longer than expected due to delays in receiving reviewer reports. The review reports acknowledge the novelty of the work; however, they also indicate that the manuscript requires major revisions before it can be considered for further evaluation.

**Journal Requirements:**

3) Your manuscript is missing the following sections: Design and Implementation, Results, and Availability and Future Directions. Please ensure that your article adheres to the standard Software article layout and order of Abstract, Introduction, Design and Implementation, Results, and Availability and Future Directions. For details on what each section should contain, see our Software article guidelines:

https://journals.plos.org/ploscompbiol/s/submission-guidelines#loc-software-submissions

5) We notice that your supplementary Tables, and information are included in the manuscript file. Please remove them and upload them with the file type 'Supporting Information'. Please ensure that each Supporting Information file has a legend listed in the manuscript after the references list.

6) Please ensure that the funders and grant numbers match between the Financial Disclosure field and the Funding Information tab in your submission form. Note that the funders must be provided in the same order in both places as well.

**Reviewers' comments:**

Reviewer's Responses to Questions

**Comments to the Authors:**

Reviewer #1: The authors of this work describe the development of a mutability pipeline for the purpose of protein design. They have opted to using ESMFold and ESM-2 as the workhorse prediction algorithms, while providing their own mutability and energetic screening algorithms.

The pipeline is innovative, and many of the energetic computations are intuitive. It is required by PLOS Comp. Biol. that this software be already wide-spread, however it is very promising to be used by wide-audience, therefore, I do not see this as solid weakness. In the appendix, authors provide sufficient details that allow reproducibility of the work. The work has very clear road map in their future outlook as well.

Both ESMFold and ESM-2 are well documented in terms of validation and accuracy, however, as a software article, results should show “Examples of biological problems solved using the software, including results obtained with the deposited test data and associated parameters.” The four applications are rather demonstrative than solid ground applications. Results showed how the energetics of the models were improved, however, the authors do not provide sufficient biological insights to how these models could be validated or used. The work can benefit greatly by validation molecular dynamics simulations of the 10 main protein examples described. One of the applications targeting IDR is available as recent conference paper (reference 33), while another is in the form of preprint (reference 39). Another application citing Reference 38 is not relevant or not clear how it is connected to the text. In my opinion, one peer-reviewed published application would have been sufficient validation.

We have to keep in mind that deep learning prediction models are built around the notion of homology as guide for accuracy. In which case a model with long chain of bad pLDDTs will give bad results no matter how much it is mutated. The authors are already aware of this as described in their future outlook section. This work should be supported by sensitivity analysis that draws the reference line for overall accuracy (not only with reference to the starting model). For example, if a starting model is really bad, then the pipeline should be able to tell the user that quality is too low to proceed. To define the threshold for that, I propose a sensitivity analysis on a dataset of structures at various homology levels to the PDB.

I have reservations towards the use of the terms states and multi-states in the paper. These terms are commonly used in the protein 3D structure community to describe multi-state proteins. Multi-state proteins require very stringent and standardized definitions related often to activity or function. In the paper, multi-state was described in the narrow meaning of mutation-based states which were used for candidate generation via optimization, and that can be a source for confusion to the reader. This would sound as promising too much on the framework of general multi-state proteins. In my understanding, optimization is a heuristic approach that scans only a limited fraction of conformational space as opposed to other deterministic methods. While multiple optimizations can serve in sampling, they are not the ideal for multi-state predictions.

The authors used several intuitive model quality and energetic evaluations (some of which are very recent, e.g. ones developed by Alphafold), however, why they have not included mainstream molecular mechanics computations of potential energy, which are widely used by researchers?

How does this pipeline compare to FoldX? The FoldX program is known for its simple mutability engine to perform sequential point mutation screening (useful in protein-protein or protein-peptide interactions), followed by energetic scoring. It is known that beyond the single point mutants, the folding becomes less predictable with more mutations, especially for peptides.

Hydrophobic residues used in HydrophobicEnergy term do not seem to belong to a familiar hydrophobicity scale. Please justify why was Tyr or Cys or Ala or Gly not included?

Minor:

Page 15: “a cheaper model” >>> “a computationally-cheaper model”

“an expensive model” >>> “a computationally-expensive model”

Reviewer #2: Jakub Lála et al introduces BAGEL, a flexible energy function-based computational tool for protein engineering, which generalizes across different design tasks (e.g., peptide binder design and multi-state selective binding). BAGEL extends some existing ideas (e.g., Monte Carlo) but conceptualize energy landscape by combining energy terms. This open-source tool is useful for protein design (probably only small-size proteins) and an effective tool for the community. There are some minor comments:

1. Authors provide some beautiful examples, however, the efficiency was not mentioned. It would be better to show the system size, CPU/GPU counts and runtime for both per step and total. Furthermore, can BAGEL design large-size protein (e.g., 300 amino acids)? If so, how does runtime and resource usage (CPU/GPU scaling) compare for such cases?

2. For peptide or epitope binder design, it would be helpful to see a comparison against other AI-based models (e.g., RFdiffusion) to test BAGEL’s accuracy and performance within the current landscape of state-of-the-art methods.

Reviewer #3: The manuscript presents a strong and innovative contribution to computational protein design. BAGEL is clearly described, conceptually well-motivated, and demonstrates impressive flexibility across diverse applications. The framework’s emphasis on modularity, non-differentiable objectives, and multi-state optimization represents a meaningful advancement over existing design pipelines. Minor revisions are recommended to improve clarity in notation, provide brief benchmarking or validation data, and expand on integration with existing tools. Overall, this is a well-executed and valuable piece of work that should be of broad interest to the protein engineering community.

**Have the authors made all data and (if applicable) computational code underlying the findings in their manuscript fully available?**

Reviewer #1: Yes

Reviewer #2: Yes

Reviewer #3: Yes

PLOS authors have the option to publish the peer review history of their article (what does this mean?). If published, this will include your full peer review and any attached files.

Reviewer #1: **Yes: **Yazan Haddad

Reviewer #2: No

Reviewer #3: No

**Figure resubmission:**
---

## [Decision Letter · Decision Letter 1]

20 Nov 2025

Dear Dr Angioletti-Uberti,

We are pleased to inform you that your manuscript 'BAGEL: Protein Engineering via Exploration of an Energy Landscape' has been provisionally accepted for publication in PLOS Computational Biology.

Best regards,

Amar Singh

Academic Editor

PLOS Computational Biology

Arne Elofsson

Section Editor

PLOS Computational Biology

Thank you for bringing your concerns about the review report from Reviewer #3. As an editor, I have informed the editorial office about your concerns. I reviewed the report, and while some comments are relevant and provide valuable feedback, other portions of the report appeared generic and inconsistent. Thus, we decided to rely on the other two review reports for further evaluation of your revised manuscript. I am pleased to inform you that the other two reviewers have approved the revised version.

Reviewer's Responses to Questions

**Comments to the Authors:**

Reviewer #1: The authors have answered all my comments diligently and I thank them for enriching the discussion. I fully recommend this article for publication.

Reviewer #2: All concerns were well addressed. I support it is published as a novel tool for protein engineering, and it will be benefit to the scientific community.

**Have the authors made all data and (if applicable) computational code underlying the findings in their manuscript fully available?**

Reviewer #1: Yes

Reviewer #2: Yes

PLOS authors have the option to publish the peer review history of their article (what does this mean?). If published, this will include your full peer review and any attached files.

Reviewer #1: **Yes: **Yazan Haddad

Reviewer #2: No

---

## [Editor Report · Acceptance letter]

PCOMPBIOL-D-25-01443R1

BAGEL: Protein Engineering via Exploration of an Energy Landscape

Dear Dr Angioletti-Uberti,

I am pleased to inform you that your manuscript has been formally accepted for publication in PLOS Computational Biology. Your manuscript is now with our production department and you will be notified of the publication date in due course.

With kind regards,

Anita Estes
